# Using Species Knowledge to Promote Pro-Environmental Attitudes? The Association among Species Knowledge, Environmental System Knowledge and Attitude towards the Environment in Secondary School Students

**DOI:** 10.3390/ani13060972

**Published:** 2023-03-07

**Authors:** Talia Härtel, Christoph Randler, Armin Baur

**Affiliations:** 1Department of Biology, Eberhard Karls University Tuebingen, 72076 Tuebingen, Germany; 2Heidelberg School of Education, 69115 Heidelberg, Germany; 3Department of Biology, Heidelberg University of Education, 69120 Heidelberg, Germany

**Keywords:** species knowledge, environmental knowledge, attitude towards the environment, education for sustainable development, biodiversity

## Abstract

**Simple Summary:**

The Earth’s biodiversity is currently declining rapidly. To counteract this loss, it is important to create awareness of nature and the environment in society. According to scientists and conservationists, the basis of such awareness is knowledge of animal and plant species. This study investigated the level of species knowledge of students and the relationship between species knowledge and environmental knowledge as well as attitudes towards the environment. The study concludes that students know more vertebrate species than invertebrate species and that high species knowledge has a positive effect on environmental knowledge and attitude towards the environment. Promoting species knowledge is therefore one way to create more awareness about biodiversity.

**Abstract:**

Scientists and conservationists suggest species knowledge as a possible starting point when it comes to creating deeper knowledge and awareness of nature, the environment, and biodiversity. The aim of this work was to analyze secondary school students’ species knowledge of vertebrates and invertebrates. This is one of the first studies that also draws on invertebrates. Furthermore, we investigated whether knowledge of species forms a basis for the formation of environmental knowledge and attitude towards the environment. For this purpose, a questionnaire on species knowledge was developed. In addition, a questionnaire was used to measure environmental system knowledge, and the 2-MEV Attitude Scale to measure attitude towards the environment. The questionnaires were completed by 103 seventh and eighth-grade (age: 12–13) students of a secondary German school (Gymnasium, highest stratification level). The students identified more vertebrates than invertebrates (50.15% vs. 36.7%). The structural equation model with the latent variables species knowledge, environmental system knowledge, and attitude towards the environment showed that species knowledge has a highly significant influence on the two other latent variables. More precisely species knowledge explained 28% of the variance in environmental systems knowledge and 17% of the variance in attitude towards the environment. This study can therefore draw attention to the relevance of teaching species knowledge in the sense of Education for Sustainable Development, in order not to promote decreasing biodiversity through dwindling species knowledge.

## 1. Introduction

To enable sustainable development as defined by the Sustainable Development Goals [1], the conservation of biodiversity is of central importance [2]. However, the diversity of species on our planet is currently declining more rapidly than at any other time in human history [3]. To counteract species extinction, it is crucial to promote sustainable behavior in our society [4]. Education for sustainable development (ESD) seems to be a central key to this [5,6]. However, there is still the question of how this can be achieved in a meaningful way.

Many studies have already shown that the correlation between environmentally relevant knowledge and the sustainable behavior of people is moderate at best [7,8,9,10]. For attitudes, the relationship to behavior appears to be stronger: Hines, Hungerford and Tomera [9] reported a correlation between attitude towards the environment/ecology and environmental behaviors in their meta-analysis. Furthermore, according to current environmental psychology and sociological studies, environmental attitudes are among the most important factors that encourage people to become involved in environmental protection [11,12]. In their Environmental Literacy Model, Roczen et al. [13] stated that students’ attitude towards the environment has a positive impact on their sustainable behavior and that attitude towards the environment is also related to environmental system knowledge [13]. To contribute to understanding and awareness of nature and the environment in education, species knowledge and the ability to identify species are considered fundamental (see, e.g., [14,15,16]). However, the extent to which species knowledge is related to environmental system knowledge or attitude towards the environment is unclear. Some scientists and conservationists see it as the basis when it comes to creating deeper knowledge and awareness towards nature and the environment or biodiversity (see, e.g., [2,17,18]). Parallel to the extinction of species, there is a so-called extinction of experience [19], which means that the interactions with nature, as well as the species knowledge of students in many countries, are rather low [20,21,22,23]. So both, species, species numbers and the possibility to experience species declines in parallel [19]. On the other hand, there is also evidence that children’s experiences with nature tend to increase [24,25]. Further, species knowledge is usually assessed based on vertebrates, and therefore, assessing species knowledge by including invertebrates is an important aspect. However, it may be different when it comes to plant species identification [26,27].

### 1.1. Literature Review

#### 1.1.1. Definition of Species Knowledge

So far, there is no uniform definition of species knowledge [28]. While some authors understand it as the recognition and correct labeling/naming of species [21,23], others define species knowledge beyond the mere knowledge of identification characteristics and extend to a deeper knowledge of ecology, distribution, and systematics of species [29].

To distinguish identification skills from the more profound knowledge about species, Hooykaas et al. [30] coined the term species literacy. Species literacy is characterized by knowledge of in-depth background information about a species, such as its place in the ecological food chain, its natural habitat, or others [31]. This in-depth species literacy represents the starting point for awareness about biodiversity [2] and thus plays an important role in conservation efforts. However, Hooykaas et al. [32] showed that simple species identification skills are a very good predictor of deeper knowledge about species. For this reason, species knowledge in this article will be understood as the recognition and correct naming of a species.

#### 1.1.2. Level of Species Knowledge among Students

Several national and international studies have dealt with the analysis of species knowledge among school students. In Germany, species knowledge has often been investigated based on bird species [23,28,33]. Nevertheless, bird species knowledge is not particularly high among schoolchildren, and depending on the study, the participants were able to recognize less than a third [28] to just below 50% [33] of the presented bird species. Comparing the percentage of correct identifications in the different vertebrate classes mammals are most frequently correctly identified, followed by amphibians, fish, birds, and reptiles in descending order [23]. This is somewhat counterintuitive since most birds are diurnal, colorful, and vocally active meanwhile most mammals are nocturnal, brownish, and mostly silent. Because of this, encountering a mammal species should naturally be more difficult compared to birds. Therefore, species knowledge is probably not related to direct experience.

The decrease in species knowledge during the last decades is less well understood [17,23]. Randler [21], found no changes in vertebrate knowledge compared to Eschenhagen [34]. However, comparing Randler [21] with Gerl, Randler and Neuhaus [17] showed a decrease in knowledge of about 15%. Referring to the most recent study on species knowledge of students from Germany, less than two-thirds of the vertebrate species are recognized [17]. For plants, species knowledge seems to be even lower [35].

In Switzerland, children and adolescents were asked about the species they noticed on their way to school. On average, these were eleven species, most frequently cats (*Felis catus*), dogs (*Canis lupus familiaris*), and “birds” [14]. Patrick et al. [36] asked students from six countries to name as many animals as possible. Mammals were the most frequently named class of animals in all countries and especially the mammals that students often encounter in everyday life were frequently named. This was followed by birds while invertebrates were rarely listed [36]. This further supports the better knowledge of mammal species compared to other taxa.

Similar to Sturm, Voigt-Heucke, Mortega and Moormann [28], Prokop and Rodák [22] conducted a study among Slovak students to investigate the students’ ability to identify 25 bird species using auditory and visual stimuli. About 19% of the birds were identified by their song, 34% by their habitus, and 45% were identified when both the acoustic and visual stimuli were provided to the students [22]. Genovart, Tavecchia, Enseñat and Laiolo [20] confirmed that students often know exotic species better than native species. Only native amphibians and reptiles were better known than the exotic species.

#### 1.1.3. Environmental Knowledge and Its Relation to Species Knowledge

Analogous to species knowledge, there is no uniform definition of environmental knowledge.

In psychological environmental research, environmental knowledge has usually been assessed based on one or even two forms of knowledge. Hines, Hungerford and Tomera [9] distinguish between two dimensions of environmental knowledge: the knowledge about environmental problems and the knowledge about the options to act on a certain environmental problem. Schahn and Holzer [37] also differentiate between abstract school knowledge and concrete action knowledge.

Environmental knowledge as a construct of three components—system knowledge, action-related knowledge, and effectiveness knowledge—seems to be gaining acceptance in current research [38] and was also used in this study.

System knowledge refers to knowledge about how ecosystems function and the existence of environmental problems [7]. For example, knowledge about the relationship between CO_2_ emissions and global warming belongs to system knowledge and is commonly referred to as “knowing what” [7]. Action-related knowledge, commonly referred to as “knowing how” (ibid.), includes knowledge about options for action and ways of acting. This involves for example knowledge of correct waste separation [39]. Effectiveness knowledge, or “knowing how to get the greatest environmental benefit” [7], is relevant to successful action. It addresses the benefit associated with a particular behavior and thus demonstrates its potential to protect the environment [40]. Effectiveness knowledge, unlike Action-related knowledge, thus requires cost-benefit considerations. For example, it may be more effective to buy a new fuel-efficient car instead of using an older car less often [7].

To what extent species knowledge can be specifically located in environmental knowledge remains unclear based on the definition. In the literature, some authors include species knowledge in their conceptual understanding of environmental knowledge, and in some cases, environmental knowledge has even been assessed by testing species knowledge [34,41]. In this study, we decided to survey species knowledge independently of environmental knowledge to be able to investigate possible relationships.

#### 1.1.4. Definition of Attitude towards the Environment and Its Relation to Species Knowledge

To date, there is no uniform definition of attitude towards the environment or even environmental awareness in the literature, as there is a need for discussion, particularly about the dimensionality of the term. Maloney and Ward [42] subdivided the construct of “attitude towards the environment” into four components, consisting of knowledge, affect, verbal commitment, and actual commitment. Other authors did not include all components in their understanding of environmental awareness [37,43] or added an ethical-normative component of environmental awareness, which is expressed in environmentally relevant values and social norms.

Most studies, however, understand environmental awareness as an attitude or attitudinal construct (see, e.g., [44]). Moreover, Roczen, Kaiser, Bogner and Wilson [13], who delineated attitude towards the environment in their Environmental Literacy Model, point out that it represents the associated motivation for a person’s sustainable lifestyle.

More specifically, following Wiseman and Bogner [44], the concept of attitude towards the environment is conceived as attitudes that can be grouped into two sets. According to Rokeach [45] sets of enduring attitudes are called values and underlie attitudes and actions. In the context of attitude towards the environment, values refer to preferences for environmental Preservation (PRE) and Utilization (UTL). Both preferences can be conceptualized as ecocentric and anthropocentric thinking, which is reflected in two distinct worldviews [44]. The anthropocentric worldview supports attitudes that favor the use and exploitation of natural resources, thereby paying homage to environmental destruction [46]. There might be some conceptual overlap between this anthropocentric view and the egocentric view described by Goodbody [31]. The ecocentric worldview, on the other hand, is characterized by living in harmony with nature and emphasizes maintaining the natural balance [47]. In summary, attitude towards the environment consists of an interplay of attitudes that can be assigned to either an ecocentric or an anthropocentric view of the world.

To what extent, or if, species knowledge and attitude towards the environment are related has not been directly investigated in studies so far. However, some studies have been conducted internationally in the past, confirming that species knowledge and attitudes towards different organisms positively influence each other [48,49]. Furthermore, higher sympathy towards certain animal species is associated with increased species knowledge, highlighting the visual attractiveness or “beauty” of an animal species as an influencing factor for positive attitudes towards it [26,48,50,51]. However, whether species knowledge is responsible for positive attitudes towards species or vice versa is not clear from most studies. Only Schlegel, Breuer and Rupf [49] assume that species knowledge forms an important basis for positive attitudes.

### 1.2. Research Aims and Questions

Therefore, the first research aim of this study was to survey the species knowledge of students in 2021 and to investigate differences in the identification of vertebrates compared to invertebrates, as there are hardly any studies on invertebrate knowledge so far. The second aim was to examine the impact of species knowledge, more precisely, whether students’ species knowledge influences their environmental systems knowledge and attitude towards the environment. If applicable, this will draw attention to the relevance of teaching species knowledge in terms of ESD.

The Research Questions of this paper are:(1)How good is students’ species knowledge?

Assessing the current state of knowledge of species among schoolchildren was an objective of this study. The focus was especially on the comparison of knowledge about vertebrates versus invertebrates. Based on the conceptual framework, invertebrate knowledge has hardly been studied so far and if so, it seems to be worse than vertebrate knowledge. To our knowledge, species knowledge surveys of invertebrate species have not yet been conducted in Germany.

(2)To what extent does students’ species knowledge influence their environmental systems knowledge and their attitudes towards the environment?

Some scientists and conservationists suggest species knowledge as a starting point to create deeper knowledge and awareness towards nature and the environment or biodiversity [2,17,18,52]. Moreover, environmental system knowledge seems to be the starting point of environmental knowledge, so it can be assumed that species knowledge is closely linked to system knowledge, as well as attitudes towards the environment.

## 2. Materials and Methods

### 2.1. Sample

Surveys were conducted in five school classes of a secondary German school located in the southwest federal state of Baden-Württemberg. The school system in Germany divides students after grade four usually into three different strains based on their cognitive skills. This study was based on the highest stratification level (Gymnasium, about 41% of the students are enrolled in this stratification in BW [53]). The questionnaires on species knowledge and attitude towards the environment were completed by 108 school students (55 female, 53 male), and the questionnaire on environmental system knowledge was completed by 103 students (55 females, 48 males). All seventh and eighth-grade students of this school participated and were between 12 and 13 years old (*M* = 12.55; *SD* = 0.58). The questionnaires were anonymous, and consent was obtained from the headmaster and the participants.

### 2.2. The Questionnaires

#### 2.2.1. Species Knowledge

The selection of the animal species for the questionnaires was based on an expert rating. Together with five experts from the Nature Protection Organization in Germany Naturschutzbund (NABU), a questionnaire was developed to assess the species knowledge of students based on vertebrates (*Vertebrata*) and invertebrates (*Invertebrata*). The NABU is the nature protection organization in Germany with the highest number of members (875.000; NABU [54]). In some studies, the representation of species using pictures has already proven successful [3,21,23] and in this study, species knowledge was also assessed using color photographs.

Although there are many more invertebrate species than vertebrate species in Germany or Baden-Württemberg [55] more vertebrates were asked in the questionnaire due to their higher popularity and familiarity among students [26,36]. Following Hollstein [56] 30 animal species were selected together with the five experts from NABU according to the following two criteria:(a)frequency and abundance of the species (especially in Baden-Württemberg)(b)the conspicuousness of the species (e.g., by size, coloration, and habitat) (following Hollstein [56])

The degree of agreement among the experts about which species should be included in the questionnaire can be seen in Table 1.

To facilitate the experts’ selection of the species, a preselection of 90 animal species was made. For this preselection, the International Union for Conservation of Nature (IUCN) Red Lists of Threatened Species of Baden-Württemberg was used to exclude endangered and therefore very rare animal species. Since amphibians (*Amphibia*) and reptiles (*Reptilia*) are endangered in general, an exception for them was made to include some species in the preselection. In addition, a popular science book from Schmid [57] was also used for the preselection, since it focuses on the 100 most common and conspicuous animal species in Germany. Furthermore, the so-called “Stunde der Gartenvögel 2021” [58] as well as the hunting statistics for 2019/2020 and the Neckar report, a report on the fish fauna of the river “Neckar” in Heidelberg, from 2008 were used [59,60]. The “Stunde der Gartenvögel” is a Citizen Science day, where the general population participates in garden bird counts (which needs a low knowledge and commitment, Randler [61]). Thus, these statistics also shed light on species that are encountered by the interested public.

The number of animal species used in the questionnaire does not correspond to the percentage of their respective number of species within a vertebrate/invertebrate class in Baden-Württemberg/Germany (see Table 2). This is because each vertebrate class should be represented with at least two species in the questionnaire to represent a diverse spectrum of animal species. Since fish (*Pisces*) are not conspicuous to most children and adolescents because they are underwater, only two instead of three fish species were integrated into the questionnaire. On the other hand, the proportion of mammals (*Mammalia*) within the vertebrates was increased, since students show a strong affinity for mammals [62]. Crustaceans (*Crustacea*) play a rather minor role in the lifeworld of students due to their habitat, which is mostly located in or near the water and therefore they were excluded from the questionnaire. To represent the diversity of the hexapods (*Hexapoda*), one species each from the order of dragonflies and damselflies (*Odonata*), grasshoppers (*Orthoptera*), hymenopterans (*Hymenoptera*), beetles (*Coleoptera*), butterflies (*Lepidoptera*) and dipterans (*Diptera*) were included in the questionnaire. In addition, the NABU experts decided to include the common earwig (*Forficula auricularia*) in the questionnaire due to its abundance and conspicuousness. Therefore earwigs (*Dermaptera*) were added to the hexapods as the seventh order. The expert rating resulted in ten invertebrates and 20 vertebrates, each represented by a representative image. The images were mainly taken from copyright-free sources and were also checked with the help of the experts to ensure that the animal was easily recognizable and that all decisive identification features were depicted. For the bird (*Aves*) species, male animals were used, as these usually have more conspicuous identification features. In addition, the specialist literature was used to make sure that all identifying features were present.

In the questionnaire on species knowledge, a maximum of 30 and a minimum of zero points could be achieved. The grading followed a partial credit model: The students received one point for naming the correct German species name (All students were able to speak, read and write the German language at least at B2-Level). For a total of eight animal species, full points were awarded for alternative names in addition to the species name (for example “Spatz” for “Haussperling” (*Passer domesticus*)), such as colloquial names or systematically higher units. If the students did not give the correct species name, but instead the genus or family name, they received 0.5 for their answer. Information of higher systematic units, such as the order name, was scored 0. Thus, if the students indicated “mallard” for *Anas platyrhynchos*, they received 1.0 points, while they received 0.5 for “duck” and zero for “waterfowl” (order Anseriformes) or “bird” (class). This differentiated partial credit scoring is often more adequate to describe the participant’s species knowledge than dichotomous scoring [23].

#### 2.2.2. Attitude towards the Environment

In addition to the species knowledge questionnaire, students were also given a questionnaire to measure their attitude towards the environment. For this purpose, the 2-MEV Attitude Scale was used, which has been developed and independently tested by Wiseman and Bogner since the 1990s (see, e.g., [64,65,66,67,68]) and is a reliable and valid measurement tool. In this study, the items from Thorn and Bogner [69] were used.

The 2-MEV-Scale consists of 2 subscales, each of which measures environmental Utilization (UTL) preferences and Preservation (PRE) preferences based on 10 items. The items were scaled using a five-point Likert scale (1 = completely true, 2 = fairly true, 3 = undecided, 4 = fairly false, and 5 = completely false).

In the attitude towards the environment questionnaire, values, therefore, ranged from 1 to 5 based on the mean value of each subscale. High values in the PRE-Subscale correspond to a strongly ecocentric attitude, while high scores in the UTL-Subscale correspond to a strongly anthropocentric attitude.

#### 2.2.3. Environmental System Knowledge

The environmental system knowledge was developed and measured by Roczen, Kaiser, Bogner and Wilson [13] and designed for use with students in sixth, seventh, or eighth grades. The questionnaire contained a total of 38 items, of which 14 questions were multiple choice questions with one correct answer, 13 questions were in true/false format and eleven questions were multiple choice questions with multiple correct answers. An example of a question in the multiple-choice format is “On clear nights, why does it get colder towards the morning?”, and a statement in true/false format is “Solar energy is unlimitedly available”.

One point was awarded for each correctly answered question in the environmental systems knowledge questionnaire and students could score between 0 and 38 points.

### 2.3. Implementation

Before the surveys were conducted, the questionnaire on species knowledge was pre-tested with four seventh-grade students who did not participate in the final survey. The surveys were carried out in September 2021. To complete the questionnaire on species knowledge, the participants were given about 15 min (corresponding to 30 s per illustration), for the questionnaire on attitude towards the environment ten minutes (corresponding to 30 s per statement) and for the questionnaire on environmental system knowledge 25 min (corresponding to 50 s per multiple choice question and 20 s per true/false question), whereby only a few students used up the time to the end. The survey was kept anonymous using codes to facilitate questionnaire matching.

### 2.4. Internal Consistency of the Questionnaires

To determine the internal consistency of the species knowledge questionnaire, Cronbach’s alpha was calculated for the 30 animal species (Cronbach’s alpha = 0.86). In knowledge tests, usually alpha levels are lower [70], however, due to the large number of items the alpha level might have become quite good. The Cronbach’s alpha for the PRE subscale was 0.59 and for the UTL subscale 0.73. Therefore, the reliability of the UTL subscale was found to be high [71], and since previous studies also continued with similarly low-reliability values of the PRE subscale [65,72], the item number was not reduced in this work and the reliability was assumed to be acceptable. Cronbach’s alpha was also calculated for the environmental systems knowledge questionnaire to determine internal consistency. This was in the moderate range at 0.59 [71]. However, since Frick [73] also continued with a Cronbach’s alpha of 0.55 during questionnaire development, reliability was also assumed to be acceptable here and no items were removed.

### 2.5. Statistical Analysis

The program SPSS 27 (IBM; Armonk, NY, USA) was used for statistical analysis of all data and additionally R version 4.0.5 (R Core Team, Vienna, Austria) was used to calculate the structural equation model (SEM).

All questionnaires were also tested for normal distribution using the Shapiro-Wilk test, graphical analysis, and z-standardization of skewness and kurtosis.

Paired *t*-tests were calculated to test whether the means of the two dependent samples differ. To compare percentages, we used non-parametric tests. As a measure of effect size, Cohen’s h (*h*) was calculated [74].

Factorial ANOVAs were used to test the influence of gender, age, or class on the dependent variables (species knowledge, environmental system knowledge, and attitude towards the environment).

Research Question 2 was analyzed by Structural Equation Modeling (SEM). The estimation of SEM entails some preconditions such as linearity and multivariate normal distribution [75] were previously confirmed. In addition, the respective subscales were also checked, and a normal distribution could be confirmed for all of them. To test for linearity, correlation matrices of all variables were computed, each belonging to a measurement model. The strength and direction of the correlation were as expected, and the conditions of linearity were met accordingly. Subsequently, the measurement models were defined and then subjected to a factor-analytical goodness of fit, also called fit. Based on this, the parameters could be estimated. The fit indices Comparative Fit Index (CFI), Goodness of Fit Index (GFI), Standardized Root Mean Square Residual (SRMR), and Root Mean Square of Error Approximation Index (RMSEA) were used to evaluate the model quality. The thresholds, in which the fits are in acceptable to good ranges, are >0.90 for CFI and GFI and <0.08 for RMSEA and SRMR [76]. The significance of the postulated relationships and the relationship between environmental system knowledge and attitude towards the environment already confirmed by Roczen, Kaiser, Bogner and Wilson [13], was determined with the calculated path weights as well as with the significance level of the parameters.

Finally, the SEM was calculated, and the latent constructs and manifest variables were modeled in their relationships according to the postulated hypotheses. Depending on the scaling of the variables, two estimation procedures were used: the Maximum Likelihood (ML) procedure for interval-scaled variables and the Diagonally Weighted Least Squares (DWLS) for ordinally scaled variables.

## 3. Results

### 3.1. How Good Is Students’ Species Knowledge?

The students were able to recognize an average of 13.70 (*SD* = 4.66; range 5.5–24.5; *mdn* = 13.5; *D* = 9.5) out of 30 animal species based on the partial credit scoring (see Figure 1). An average of 10.03 (*SD* = 3.30; *mdn* = 10; *D* = 7.5) out of 20 vertebrates and 3.67 (*SD* = 1.65; *mdn* = 3.5; *D* = 5) out of ten invertebrates were correctly named.

Based on the species-level scoring without applying for the partial credits, the mean was 11.41 (*SD* = 4.67; range 4–22). On average, 8.34 (*SD* = 3.22) vertebrates and 3.06 (*SD* = 1.82) invertebrates were correctly named with the species’ name. No significant differences in gender, age, and class of the students were significant (one-way ANOVA: gender: F(1,106) = 0.548, *p* = 0.461; age: F(3,104) = 0.558, *p* = 0.644; class: F(4,103) = 1.240, *p* = 0.299). More than 75% of the students were able to identify the red squirrel (*Sciurus vulgaris*), wild boar (*Sus scrofa*), European hedgehog (*Erinaceus europaeus*), roe deer (*Capreolus capreolus*), and common earthworm (*Lumbricus terrestris*) at the species level. The 13 least known species (identification rate below 25%) included five of the ten invertebrates and eight out of 20 vertebrates (see Table 3). Large differences in identification rates became apparent in the distinction between genus and species levels (see Figure 2). The white stork (*Ciconia ciconia*), for example, was named by 83.33% of the students as “stork” (genus level), but only by 1.85% at the species level. The southern hawker (*Aeshna cyanea*) was not named correctly by any of the participants, as it was mostly named as “dragonfly” only at the order level.

The *t*-test showed that for the identification rates of vertebrates and invertebrates highly significant differences existed with small effect sizes (t(107) = 11.826, *p* < 0.001, *h* = 0.273) with significantly more vertebrates than invertebrates being identified by the students.

### 3.2. To What Extent Does Students’ Species Knowledge Influence Their Environmental Systems Knowledge and Their Attitudes towards The Environment?

For the SEM, the three measurement models species knowledge, environmental system knowledge, and attitude towards the environment were defined and initially tested for their fit indices. According to Zinnbauer and Eberl [76] the measurement model of species knowledge, consisting of the six subscales of knowledge of invertebrates, knowledge of birds, knowledge of mammals, knowledge of fish, knowledge of amphibians, and knowledge of reptiles, had a very good model fit (χ^2^(9) = 9.537, *p* = 0.389, *CFI* = 0.996, *GFI* = 0.97, *SRMR* = 0.035, *RMSEA* = 0.024). Bird species knowledge and invertebrate knowledge had the highest effect sizes (*β* = 0.84 and *β* = 0.81) compared to the other subscales of species knowledge (see Figure 3). The measurement model of environmental system knowledge consisted of three subscales: system knowledge 1 (consisting of the multiple-choice questions with one correct answer (items 1–14)), system knowledge 2 (consisting of the true/false questions (items 15–27)) and system knowledge 3 (consisting of the multiple choice questions with multiple correct answers (items 28–38)). The model fit of this measurement model also proved to be very good (χ^2^(3) = 23.916, *p* = 0.287, *CFI* = 0.999, *GFI* = 0.999, *SRMR* < 0.001, *RMSEA* < 0.001) [76]. The measurement model of attitude towards the environment derived from the PRE- and UTL-Subscale could not be tested for its model fit because the model was under-identified. Nevertheless, the entire SEM could be calculated because sufficient overidentified measurement models were included and thus enough known parameters could be achieved in the entire model.

Fit indices in the postulated model were also found to be very good (χ^2^(41) = 29.360, *p* = 0.913, *CFI* = 0.999, *GFI* = 0.950, *SRMR* < 0.001, *RMSEA* < 0.001) [76]. Looking at the paths in the model, species knowledge was found to be a highly significant determinant for both environmental systems knowledge and attitude towards the environment (environmental systems knowledge: *p* = 0.003, attitude towards the environment: *p* = 0.004). Species knowledge explained 28% of the variance in environmental systems knowledge and 17% of the variance in attitude towards the environment (environmental systems knowledge: β^2^ = 0.53^2^ = 0.28; attitude towards the environment: β^2^ = 0.41^2^ = 0.17). Environmental system knowledge and attitude towards the environment were significantly positively correlated (*p* = 0.020). This means that the more environmentally aware students are, the more knowledge they acquire about the environment, or vice versa: the more students know about the environment, the higher their appreciation of the environment.

## 4. Discussion

Using an example based on 30 frequently occurring, abundant, and conspicuous animal species in the living environment of German schoolchildren, this study showed that the participants’ knowledge of species is quite low. On the species level, only about one-third of the species were named correctly and based on the partial credit model, this was less than half of the animal species used in the questionnaire.

At the Gymnasium, zoological knowledge is taught primarily in the fifth and sixth grades and biology lessons take place here for two or three hours per week [77]. In grade 7, 2 h of biology per week, and in grade 8, one hour of biology per week follow. However, there was a large variance among the students, which requests further studies to explain. This may be because species learning was experienced differently in the different primary schools of the students before coming together in secondary school. On the other hand, school education may only be partly dependent on species knowledge. A short distance of the students’ home to nature and parents or other caregivers who themselves have a high level of species knowledge positively influence the species knowledge of children and adolescents [30,78,79,80,81]. In addition, recreational activities such as visits to zoos or parks also increase the species knowledge of individuals [81,82]. Another reason for the wide variation in species knowledge could be the interest of children and adolescents. In early childhood, the “universal” interest dominates, which means that children are interested in almost everything by themselves. With age, this interest becomes more specified [83]. While some secondary school students have a high interest in (field) biological topics, another part of the students are interested in other topics and therefore probably score lower on the test [23].

Mammals were the vertebrate class with the highest proportion of correctly identified species in this study. The low strength of the influence of mammal knowledge on species knowledge in the SEM is probably the result of this: since there was not enough variance in the results in mammal knowledge between the participants, it is less suitable for comparative studies of species knowledge. The result that mammals are the best-known animals is also consistent with other studies on species knowledge [21,23,30,34] and can be justified by the fact that mammals are frequently featured in the media and are the basis of many children’s toys and books [36]. Furthermore, children and adolescents generally associate animals with mammals [84] and our own phylogenetic relationship as a mammal species. The experience of mammals in nature may not necessarily be the reason, because in comparison to birds, most mammals are nocturnal or at least crepuscular [85], often dull-colored and less vocal compared for example to birds. So, birds are most easily encountered but less well-known.

Some species, such as the common toad, red fox, honeybee, and great tit were correctly identified at the genus or family level by most of the students, but only a few identified them at the species level. Gerl, Randler and Neuhaus [17] gave two reasons for this: on the one hand, it could be that the students are aware of the systematics and know that, for example, toads could be identified more specifically, but they did not know the correct name. On the other hand, it is possible that the students are not aware of the systematics and named “toad”, believing that they have identified the animal exactly. This could be studied further by using interviews and qualitative analyses.

In this study, the students’ knowledge of vertebrates is significantly higher than their knowledge of invertebrates. Possible reasons for this are for example the lower perception of invertebrates in the environment [86] since they are smaller and students are less likely to form an emotional attachment to invertebrates [26]. Consequently, fewer direct nature observations of invertebrates occur, which are important for acquiring species knowledge [87,88,89]. However, the observance of mammals such as badgers and foxes in real life is rare as well, but students’ perception of their existence seems to be much higher than that of invertebrates. In contrast to vertebrates, invertebrates play a rather minor role in the media and are also less represented in (children’s) books. In addition, the lower species knowledge of invertebrates may be related to the fact that they are often referred to by their order name in the general population. Furthermore, invertebrates more often are considered disgusting and provoke fear [90,91]. Ultimately, students do not seem to be as interested in invertebrates as they are in vertebrates [92]. However, interest facilitates learning and encourages individuals to take a closer look at the subject of learning [81] and is therefore relevant for species knowledge.

The main associations examined in the SEM were the influence of species knowledge on environmental system knowledge and attitude towards the environment. Both associations were statistically significant (*p* < 0.01). Thus, it can be confirmed that species knowledge influences both environmental system knowledge and attitude towards the environment in students. As it is based on a SEM, we can assume causality. Consequently, species knowledge as the basis of knowledge and attitudes regarding the environment no longer remains a conjecture of conservationists and scientists [2,17,18]. The influence of species knowledge on environmental system knowledge is greater than the influence on attitude towards the environment, explaining nearly 30% of the variance in system knowledge compared to 17% of the variance in attitude towards the environment. This is probably because both the species knowledge and the environmental knowledge questionnaires ask for knowledge, whereas the other questionnaire asks for attitudes and knowledge domains may be more correlated with each other.

Nevertheless, high species knowledge has a positive effect on children’s and adolescents’ attitudes towards the environment. In addition, the positive relationship between environmental system knowledge and attitude towards the environment, already detected by Roczen, Kaiser, Bogner and Wilson [13], was confirmed. Other studies have also already confirmed this relationship [93,94]. Even mediated by environmental system knowledge, species knowledge seems to have a positive effect on attitude towards the environment or via the link between attitude towards the environment and system knowledge.

Although these relationships have not yet been investigated in any study, the great influence of species knowledge is rather less surprising. As described in the introduction, attitudes towards certain animal species have so far been associated with species knowledge [48,49]. This is also true for plants [27]. However, there are some studies that already hint at the connection between species knowledge, environmental knowledge, and attitude towards the environment which is confirmed in this paper. In a Finnish study, for example, it was shown that familiarity with species has a positive effect on willingness to donate to environmental protection [95]. In the Netherlands, Hooykaas, Schilthuizen, Albers and Smeets [32] also found that correct identification of species serves as a predictor for deeper knowledge about species. According to Andrade Salazar et al. [96], there is a positive correlation between pro-environmental attitudes and biodiversity knowledge among schoolchildren in Colombia.

This work highlights the importance of species knowledge on students’ environmental system knowledge and attitude towards the environment. According to Roczen, Kaiser, Bogner and Wilson [13], attitude towards the environment significantly determines sustainable behavior in students. In addition, sustainable behavior is necessary when it comes to protecting the biodiversity of our planet and is more important than ever due to the current massive species extinction [3,97]. Nevertheless, many studies show that there is a gap between attitudes and behavior, especially regarding the environment and that environmentally conscious attitudes do not necessarily result in the same kind of behavior [98,99,100].

Species knowledge can be surveyed very well using birds and invertebrates, respectively. Since bird species knowledge and invertebrate knowledge had the highest effect sizes with respect to species knowledge. This might be because these subscales differentiate more between the individual students. After all, there is only a low differentiation within the mammal subscale, i.e., most students know the fox. This also underpins the methodology of many European species knowledge studies that relate to bird species [19,79,101,102]. Furthermore, it highlights the importance of bird species and invertebrate knowledge, because both are suitable to improve overall species knowledge.

Consequently, a greater focus should be placed on the acquisition of species knowledge in schools in the future. For this purpose, didactic methods should be developed that promote the acquisition of species knowledge in the best possible way, especially regarding birds and invertebrates [16]. In the context of the low invertebrate knowledge of students, concepts should be developed that can reduce aversions towards invertebrates and increase interest [90]. Evaluating the effectiveness of such methods and concepts to provide them to biology teachers should be the aim of future research. However, this also requires that curricula be designed more for the acquisition of species knowledge, especially for the teaching of invertebrate species and also needs to improve teacher education. However, it is important to mention that in terms of ESD, a holistic, pluralistic, and action-oriented approach should be taken [103]. On the one hand, more emphasis should be placed on the teaching of species knowledge, but this should be embedded in an action-oriented framework since ESD should motivate students to act sustainably in the sense of the Sustainable Development Goals [104].

### Limitations

Further research would also be needed to find out which factors, besides species knowledge, affect environmental system knowledge and attitude towards the environment. A limitation of this study is that only environmental system knowledge was examined, not action-related and effectiveness knowledge. In future studies, it would be interesting to investigate these two additional types of knowledge, as well as the sustainable behavior of students and the influence of species knowledge on them. In addition, it would be advisable to survey other socio-demographic factors besides age and gender, such as parents’ educational attainment, the proximity of residence to nature, ownership of a garden [79], etc., in order to examine which other factors influence the students’ species knowledge or environmental system knowledge and attitude towards the environment. Measuring attitudes with other scales that include more than two sets of attitudes e.g., [105]) would also be an interesting comparison for the future. Because of the above-mentioned environmental attitude-behavior gap, researching environmentally friendly behavior in the context of species knowledge, environmental knowledge and attitudes towards the environment would be another goal of future studies. The main limitation of the survey is the small sample size and that representativeness was not ensured as only one school participated in the survey. This prevents us from generalizing conclusions for the total population of students in Germany. Therefore, the study should be extended to other schools.

## 5. Conclusions

In conclusion, this study highlights the importance of species knowledge. The species knowledge of students is rather low, but it affects both environmental system knowledge and the attitude towards the environment. In particular, more emphasis should be placed on improving invertebrate knowledge and bird species knowledge, because the promotion of species knowledge in school is a measure to improve the knowledge and attitudes of students regarding the environment. Furthermore, this creates awareness regarding the conservation of biodiversity as required in Article 13 of the Convention on Biological Diversity (CBD) [97]. Consequently, the teaching of species knowledge also plays a role in the context of ESD and is a first step in the protection of biodiversity and the achievement of the Sustainable Development Goals [1].

## Figures and Tables

**Figure 1 animals-13-00972-f001:**
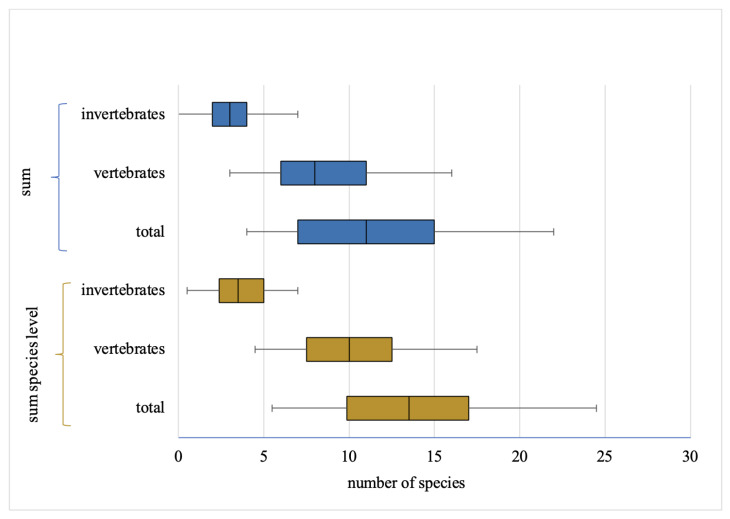
Boxplots of the achieved sum (inclusive genus/family level) and the sum on species level (both in total and for vertebrates/invertebrates) in the species knowledge test.

**Figure 2 animals-13-00972-f002:**
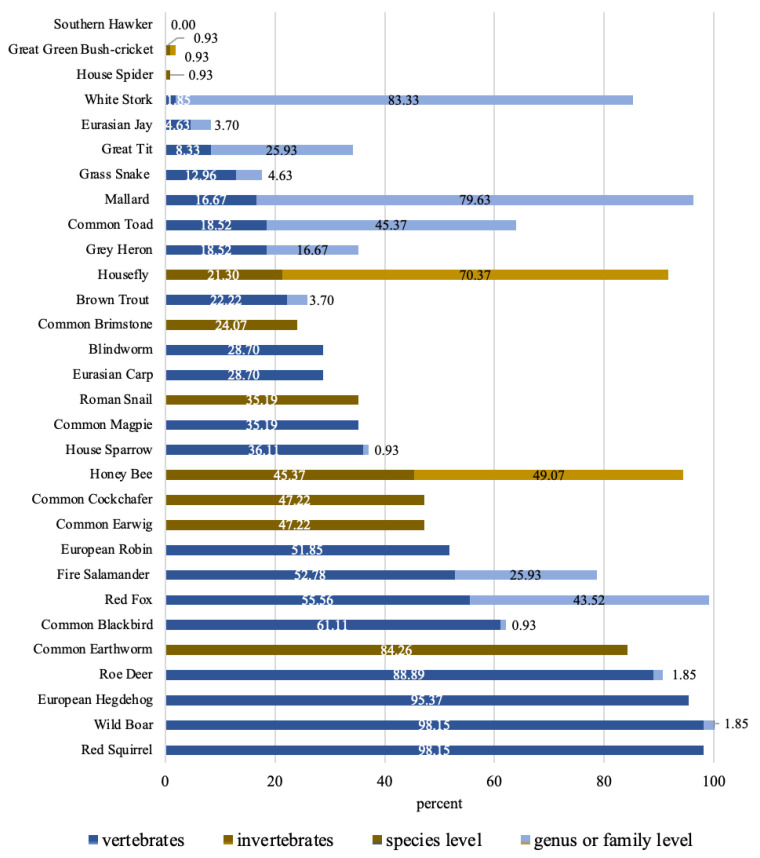
Identification rates of the animal species; blue: vertebrates, gold: invertebrates; light: identification on genus/family level; dark: identification on species level; *N* = 108.

**Figure 3 animals-13-00972-f003:**
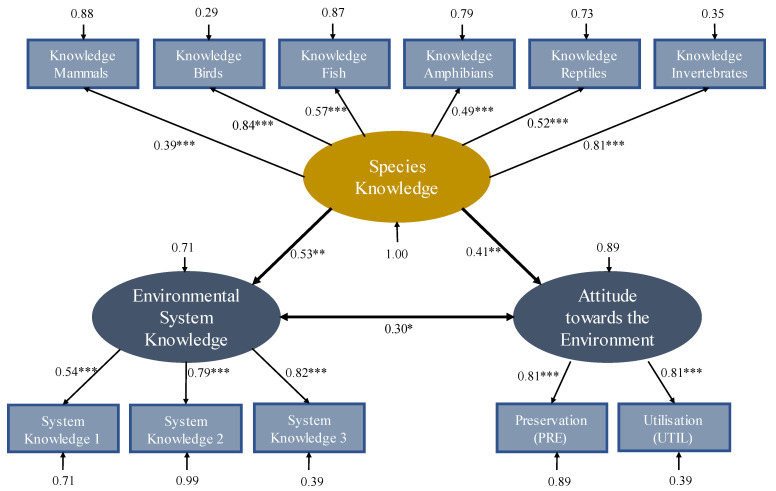
The influence of species knowledge on environmental system knowledge and attitude towards the environment among students. Notes: squared boxes represent manifest variables; oval boxes represent latent variables. Numerical values indicate the standardized multiple regression coefficients (β-values) or Pearson correlation coefficients (see double arrow). These coefficients describe the strength of the influence. Arrows without an origin denote the measurement error proportion or the unexplained variance proportion. *** *p* < 0.001; ** *p* < 0.01; * *p* < 0.05; *N* = 103.

**Table 1 animals-13-00972-t001:** Degree of agreement of the expert opinion on which species should be included in the questionnaire. The common English name, the scientific name as well as the class and order of the respective species are given.

Species	Scientific Name	Class: Order	Degree of Agreement (%)
House Sparrow	*Passer domesticus*	Aves: Passeriformes	60
Common Blackbird	*Turdus merula*	Aves: Passeriformes	80
Great Tit	*Parus major*	Aves: Passeriformes	100
Common Magpie	*Pica pica*	Aves: Passeriformes	80
European Robin	*Erithacus rubecula*	Aves: Passeriformes	100
Eurasian Jay	*Garrulus glandarius*	Aves: Passeriformes	60
White Stork	*Ciconia ciconia*	Aves: Ciconiiformes	100
Grey Heron	*Ardea cinerea*	Aves: Pelecaniformes	60
Mallard	*Anas platyrhynchos*	Aves: Anseriformes	80
Roe Deer	*Capreolus capreolus*	Mammalia: Artiodactyla	100
Wild Boar	*Sus scrofa*	Mammalia: Artiodactyla	100
Red Fox	*Vulpes vulpes*	Mammalia: Carnivora	100
Red Squirrel	*Sciurus vulgaris*	Mammalia: Rodentia	100
European Hedgehog	*Erinaceus europaeus*	Mammalia: Eulipotyphla	100
Brown Trout	*Salmo trutta fario*	Actinopterygii: Salmoniformes	100
Eurasian Carp	*Cyprinus carpio*	Actinopterygii: Cypriniformes	100
Common Toad	*Bufo bufo*	Amphibia: Anura	100
Fire Salamander	*Salamandra salamandra*	Amphibia: Urodela	100
Blindworm	*Anguis fragilis*	Reptilia: Squamata	100
Grass Snake	*Natrix natrix*	Reptilia: Squamata	100
House Spider	*Tegenaria domestica*	Arachnida: Araneae	80
Southern Hawker	*Aeshna cyanea*	Insecta: Odonata	100
Great Green Bush-cricket	*Tettigonia viridissima*	Insecta: Orthoptera	80
Honey Bee	*Apis mellifera*	Insecta: Hymenoptera	80
Common Cockchafer	*Melolontha melolontha*	Insecta: Coleoptera	100
Common Brimstone	*Gonepteryx rhamni*	Insecta: Lepidoptera	100
Housefly	*Musca domestica*	Insecta: Diptera	100
Common Earwig	*Forficula auricularia*	Insecta: Dermaptera	80
Roman Snail	*Helix pomatia*	Gastropoda: Stylommatophora	80
Common Earthworm	*Lumbricus terrestris*	Clitellata: Opisthopora	100

**Table 2 animals-13-00972-t002:** Species numbers of vertebrates and invertebrates in Baden-Württemberg and Germany. The empirical probability of the vertebrate/invertebrate classes in BW/Germany as well as their number in the questionnaire is also given.

Vertebrates	Number of Species in Baden-Württemberg ^1^	Empirical Probability	Number of Species in the Questionnaire
Birds	260	0.61	9
Mammals	77	0.18	5
Fish	59	0.14	2
Reptiles	19	0.04	2
Amphibians	11	0.03	2
total	426		20
**Invertebrates**	**Number of Species in Germany ^2^**	**Empirical Probability**	**Number of Species** **in the Questionnaire**
Hexapoda	>33.305	0.75	7
Chelicerata	>3.783	0.09	1
Crustacea	>1.067	0.02	0
Mollusca	635	0.01	1
other Invertebrates	>5.328	0.13	1
total	44.118		10

^1^ LUBW [63]. ^2^ Emde, Jessel, Schedlbauer and Wolf [55].

**Table 3 animals-13-00972-t003:** The number of students (in%) who were able to identify the respective species at the species level.

Identification Rate	Vertebrates	Invertebrates
<25%	White Stork, Eurasian Jay, Great Tit, Grass Snake, Mallard, Common Toad, Grey Heron, Brown Trout	Southern Hawker, Great Green Bush-cricket, House spider, Housefly, Common Brimstone
25–50%	Blindworm, Eurasian Carp, Roman Snail, Common Magpie, House Sparrow	Honey Bee, Common Cockchafer, Earwig
50–75%	European Robin, Fire Salamander, Red Fox, Common Blackbird	-
>75%	Roe Deer, European Hedgehog, Wild Boar, Red Squirrel	Common Earthworm

## Data Availability

The data presented in this study are available on request from the corresponding author.

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
