# Peer review of "Using Species Knowledge to Promote Pro-Environmental Attitudes? The Association among Species Knowledge, Environmental System Knowledge and Attitude towards the Environment in Secondary School Students"

_animals, 2023, doi:10.3390/ani13060972_

Round 1

Reviewer 1 Report

Dear authors, it was pleasure to read the paper, however, some changes are proposed.

Abstract

L 29-30. Add approximative age range of the students. Make cler if this is upper (ISCED 3) or lower secondary (ISCED 2) school level. Also information if they were in gymnasia programme can be of help.

L 31. Be more specific. Eg. Students were able to inentify x% of invertebrates and y% of vertebrates.

L 31: delete in questionnaire.

L 32. List latent constructs in the model, and report negative outcomes as well.

Introduction

L 41. The abbreviation SDGs was introduced, however, later is used only on two palces in the text. Additionally the term should be referenced.

L 43. every behaviour impacting environment is ecological by definition of ecology. However not any action is sustainable.

L 64. I would like to mention the article, where some findings are contradicting the mantra of general loose of experiences with nature among children.

Novotný, P., Zimová, E., Mazouchová, A., & Šorgo, A. (2021). Are children actually losing contact with nature, or is it that their experiences differ from those of 120 years ago?. Environment and Behavior53(9), 931-952.

L67. only among animals, because plant identification is maybe even more common activity in schools.

L73. see previous remark.

L 78. The statement is probaly valid also for plants, fungi etc.

L179. What about egocentric views and actions? E.g. Goodbody, A. (2006). From egocentrism to ecocentrism: nature and morality in German writing in the 1980s. In Nature in Literary and Cultural Studies (pp. 393-414). Brill.

Materials and methods

L 207. Sample. Insert some of this information in the abstract.

The paragraph about sampling procedure is missing. The sample is convenient, so the rational to include students from these schools must be provided. If not before in the limitations section.

Because children were included in a study at least a sentence about anonimity, data handling, allowance of parents, etc. should be provided (GPDR).

Table 1. Scientific names of the species should be italized.

L. 327 In very diverse knowledge tests higher alphas can be regarded more as a miracle than a rule.

L 346. To compare percentages (rates) effect size is Cohen's h.

L. 362. I propose reporting of SRMR as well.

Results

In all results. Abbreviations denoting statistical values such as p, should be italized.

L375. I propose reporting median and mode, as well.

L 384. Species names of e.g. Sciurus vulgaris etc. should be italized.

Discussion.

L 464. You wrote that mammals were identified the most often. How can you explain that in the model a path coefficient between knowledge of animals and a corresponding latent variable is the lowest

L 559. Put all limitations in a separate paragraph, because there is more than only representativeness of the sample. For example number of participants included in a model is below the all proposed thresholds (Kline, 2010; Byrne, 2016).

Author Response

The content of the attached file:

Comments and Suggestions for Authors

Dear authors, it was pleasure to read the paper, however, some changes are proposed.

Dear reviewer, thank you very much for the positive feedback and your helpful comments! I have tried to implement them as best I can and hope everything is to your satisfaction.

Abstract

L 29-30. Add approximative age range of the students. Make cler if this is upper (ISCED 3) or lower secondary (ISCED 2) school level. Also information if they were in gymnasia programme can be of help.

Thanks for the advice! I added the age and explained that the school was a Gymnasium, which represents the highest stratification level in Germany.

L 31. Be more specific. Eg. Students were able to inentify x% of invertebrates and y% of vertebrates.

I added this.

L 31: delete in questionnaire.

Done.

L 32. List latent constructs in the model, and report negative outcomes as well.

Thank you! I have now described the SEM and results more precisely in lines 32-36 and hope that this is what you expected. Since the main paths were all correlated in the SEM, there are no "negative" results so far.

Introduction

L 41. The abbreviation SDGs was introduced, however, later is used only on two palces in the text. Additionally the term should be referenced.

Thank you for the hint. I have removed the abbreviation and added the corresponding reference.

L 43. every behaviour impacting environment is ecological by definition of ecology. However not any action is sustainable.

 Thank you very much, you are right! I now only write about sustainable behaviour in the manuscript.

L 64. I would like to mention the article, where some findings are contradicting the mantra of general loose of experiences with nature among children.

Novotný, P., Zimová, E., Mazouchová, A., & Šorgo, A. (2021). Are children actually losing contact with nature, or is it that their experiences differ from those of 120 years ago?. Environment and Behavior53(9), 931-952.

Thank you, I have mentioned this contradiction in the text and included the reference (see line 70-71). I have also cited the review by Soga and Gaston (2023), which confirms that human disconnection from nature is not a universal phenomenon.

L67. only among animals, because plant identification is maybe even more common activity in schools.

Thank you, we have added this comment “However, it may be different when it comes to plant species identification (Kubiatko et al., 2021; Lindemann-Matthies, 2005)” (see line 73-74)

L73. see previous remark.

L 78. The statement is probaly valid also for plants, fungi etc.

 You are right, I put it in more general terms.

L179. What about egocentric views and actions? E.g. Goodbody, A. (2006). From egocentrism to ecocentrism: nature and morality in German writing in the 1980s. In Nature in Literary and Cultural Studies (pp. 393-414). Brill.

Thank you very much! Attitude towards environment is indeed a construct that has been studied in many different ways. I mainly referred to Wiseman and Bogner's two-limb construct, as I also use this in the study. However, the egocentric view also fits very well with the anthropocentric view described in the manuscript, that fits very well, which is why I cited the reference you provided (see line 183-184).

Materials and methods

L 207. Sample. Insert some of this information in the abstract.

Done (see above).

The paragraph about sampling procedure is missing. The sample is convenient, so the rational to include students from these schools must be provided. If not before in the limitations section.

Thanks for the advice! I have now added that all seventh and eighth grade students of the school participated (see line 234-235). Furthermore, I described in the Limitations that the sample size was small, and the representativeness was not ensured since only one school participated in the survey.

Because children were included in a study at least a sentence about anonimity, data handling, allowance of parents, etc. should be provided (GPDR).

I have added: The questionnaires were anonymous, and consent was obtained from the headmaster and the participants.

Table 1. Scientific names of the species should be italized.

 Done.

  1. 327 In very diverse knowledge tests higher alphas can be regarded more as a miracle than a rule.

I have added: In knowledge tests, usually alpha levels are lower (Stadler et al., 2021) however, due to the large number of items the alpha level might have become quite good.

L 346. To compare percentages (rates) effect size is Cohen's h.

Thank you for pointing this out, I have now changed it.

  1. 362. I propose reporting of SRMR as well.

Done.

Results

In all results. Abbreviations denoting statistical values such as p, should be italized.

 Done for SD, mdn, D, p, β, CFI, GFI, RMSEA.

L375. I propose reporting median and mode, as well.

Thank you, I have added the values.

L 384. Species names of e.g. Sciurus vulgaris etc. should be italized.

 Done.

Discussion.

L 464. You wrote that mammals were identified the most often. How can you explain that in the model a path coefficient between knowledge of animals and a corresponding latent variable is the lowest

Thank you, we have added this comment: “The low strength of the influence of mammal knowledge on species knowledge in general is probably the result of this: since there was not enough variance in the results in mammal knowledge between the participants, it is less suitable for comparative studies of species knowledge.” (see line 501-505)

L 559. Put all limitations in a separate paragraph, because there is more than only representativeness of the sample. For example number of participants included in a model is below the all proposed thresholds (Kline, 2010; Byrne, 2016).

Thank you, I have added the limitations in an extra paragraph and described the problem of the small sample size.

Reviewer 2 Report

The manuscript has got an empirical character, where authors used quantitative approach toward obtaining and analyzing data. The manuscript is written in understandable form, the used statistical methods and techniques are adequate. Authors bring new ideas in the biology and also in didactics biology.

I have got only two technical comments.

1 Maybe, it would be helpful to use research aims, and they should be placed before research questions in the text. Research questions are ok.

2. Please follow guidelines for authors in references.

I hope my comments are helpful.

Author Response

The content of the attachment:

Comments and Suggestions for Authors

The manuscript has got an empirical character, where authors used quantitative approach toward obtaining and analyzing data. The manuscript is written in understandable form, the used statistical methods and techniques are adequate. Authors bring new ideas in the biology and also in didactics biology.

Thank you very much for your positive feedback! I am very pleased that you like the manuscript.

I have got only two technical comments.

1 Maybe, it would be helpful to use research aims, and they should be placed before research questions in the text. Research questions are ok.

Many thanks for the tip! I have taken your advice and added a paragraph on Research Aims and Questions (1.2).

  1. Please follow guidelines for authors in references.

Thank you very much, I have now used the template from MDPI for Endnote and hope the references are now all according to the guidelines

I hope my comments are helpful.

Reviewer 3 Report

Dear Authors,

I have some remarks that I describe line by line:

2-5 - title shoud be changed, especially its first part: " Using species knowledge to counteract species extinction?" - it is not a clue of your article. You wrote a few sentences about species extinction but only in literature review, I have impression that you put it into the title to catch people eyes, like climate change, etc. You did not research on how using species knowledge counteract species extinction. It is not THIS paper.

83 - technical issue: you wrote: Hooykaas, et al. [26], instead of Hooykaas at el [26]. "et" in Latin is like "and", you don't need coma here.

Change here and in many other places in the text.

Line 83 - I would delete [26] here, because you put it in 85 line.

87 & 88 - twice [27] is too many. One is enough.

211 - you opened '(' but did not close it in 212 line.

213 & 215 - rather females, males, than female, male

279 - in Table 1 you put names of column in big letters, here now. Why? Please standardize it.

From 384 - Latin names of species should be written in Italics

395 - Table 3 - Identification rate. Third column, 4th line - empty box should not be here. '-' or 'no data'?

Issues to discuss and think about in the context of that paper, especially chapter Discussion:

a) In general obtained results did not suprise me. Did you expect different results?

b) There is lack information about:

- details data about how many hours of biology, and even more details: hours of lessons only about vertebrates and invertebrates children could attend before the survey. I have no idea how it is in Germany but I'm very interested in such information. It can mainly affect the results.

c) did you prepared this paper only for the paper or some university schedule or would you like to show your results to German Ministry?

d) I feel a big lack of information of the similar research and results in different European countries as a background and point to discussion of the paper.

Good luck

Author Response

The content of the attachment:

Dear Authors,

I have some remarks that I describe line by line:

2-5 - title shoud be changed, especially its first part: " Using species knowledge to counteract species extinction?" - it is not a clue of your article. You wrote a few sentences about species extinction but only in literature review, I have impression that you put it into the title to catch people eyes, like climate change, etc. You did not research on how using species knowledge counteract species extinction. It is not THIS paper.

Thank you for this clarification! I have now amended the title and hope it is now to your satisfaction.

83 - technical issue: you wrote: Hooykaas, et al. [26], instead of Hooykaas at el [26]. "et" in Latin is like "and", you don't need coma here.

Change here and in many other places in the text.

Thank you for pointing this out, I have now updated it.

Line 83 - I would delete [26] here, because you put it in 85 line.

Done.

87 & 88 - twice [27] is too many. One is enough.

Done.

211 - you opened '(' but did not close it in 212 line.

Done.

213 & 215 - rather females, males, than female, male

Done.

279 - in Table 1 you put names of column in big letters, here now. Why? Please standardize it.

Thanks for the hint, I have now standardized it.

From 384 - Latin names of species should be written in Italics

Done.

395 - Table 3 - Identification rate. Third column, 4th line - empty box should not be here. '-' or 'no data'?

In fact, there is no data for this identification rate. I have now added a line to clarify this.

Issues to discuss and think about in the context of that paper, especially chapter Discussion:

  1. In general obtained results did not suprise me. Did you expect different results?

Thank you for your comment! The results have indeed not surprised us either, which hopefully becomes clear from the introduction and now also from the discussion.

  1. There is lack information about:

- details data about how many hours of biology, and even more details: hours of lessons only about vertebrates and invertebrates children could attend before the survey. I have no idea how it is in Germany but I'm very interested in such information. It can mainly affect the results.

Thank you very much, good idea to add this! I hope the new passage on this in the discussion is now to your satisfaction (see line: 470-472)

  1. did you prepared this paper only for the paper or some university schedule or would you like to show your results to German Ministry?

The results are not to be explicitly presented to the ministry, but the open source format allows politicians to access them as well. The study will eventually also be presented on the website LEAD.Schule, where a lot of information from the Tübingen research is published

  1. I feel a big lack of information of the similar research and results in different European countries as a background and point to discussion of the paper.

Through the newly inserted paragraphs in the discussion, I very much hope that I have succeeded in providing you with even more information about the study situation. Nevertheless, I would like to point out again that there are no studies so far that relate species knowledge to environmental knowledge and attitude towards environment.

Good luck

Round 2

Reviewer 3 Report

Dear Authors

Thank you for taking into account my remarks. I am satisfied with the corrected parts. Just one new issue:

In line 110 you wrote: However, comparing Randler [22] with Gerl, Randler and Neuhaus 110 [17]...

I would rather write: However, comparing Randler et al. [22]... you don't need to write surname of each author... references are enough to this purpose.

The same in other places, eg. 126, 133, 169 lines, etc.

Well done!

Author Response

The content of the attachment.

Dear Authors

Thank you for taking into account my remarks. I am satisfied with the corrected parts. Just one new issue:

In line 110 you wrote: However, comparing Randler [22] with Gerl, Randler and Neuhaus 110 [17]...

I would rather write: However, comparing Randler et al. [22]... you don't need to write surname of each author... references are enough to this purpose.

The same in other places, eg. 126, 133, 169 lines, etc.

Well done!

Dear Reviewer,

thank you very much for the positive feedback, I am very pleased that everything is to your satisfaction.

Regarding your comments on the references: I actually follow the endnote script provided by Animals. However, I am also irritated that more than 2 surnames are sometimes listed. Before making the change, I will contact the editor to ask how this is done in the Animals Journal. Thanks for the advice!
